

# The Moon as a photometric calibration standard for microwave sounders

Martin Burgdorf[1], Stefan A. Buehler[1], Imke Hans[1], Theresa Lang[1], and
Simon Michel[1]

[1]Meteorologisches Institut, Universität Hamburg, Bundesstraße 55, 20146 Hamburg, Germany

*Correspondence to:* M. Burgdorf (martin.burgdorf@uni-hamburg.de)

**Abstract.** Instruments on satellites for Earth observation on polar orbits usually employ a two-point calibration technique, in which deep space and an on-board calibration target provide two reference flux levels. As the direction of the deep space view is in general close to the celestial equator, the Moon moves sometimes through the field of view and introduces an unwelcome additional signal.

One can take advantage of this intrusion, however, by using the Moon as a third flux standard, and this has actually been done for checking the lifetime stability of sensors operating at visible wavelengths. As the disk-integrated thermal emission of the Moon is less well known than its reflected sunlight, this concept can in the microwave range only be used for stability checks and intercalibration. An estimate of the frequency of appearances of the Moon in the deep space view, a description

of the limiting factors of the measurement accuracy and models of the Moon's brightness, as well as a discussion of the benefits from complementing the naturally occurring appearances of the Moon with dedicated spacecraft maneuvers show that it would be possible to detect photometric lifetime drifts of a few percent with just two measurements. The pointing accuracy is the most crucial factor for the value of this method. Planning such observations in advance would be particularly beneficial,

because it allows to observe the Moon at well-defined phase angles and to put it at the center of the field of view. A constant phase angle eliminates the need for a model of the Moon's brightness when checking the stability of an instrument. With increasing spatial resolution of future microwave sensors another question arises, viz. to what extent foreground emission from objects other than the Moon will contaminate the flux entering the deep space view, which is supposed to originate exclu-

sively in the cosmic microwave background. We conclude that even the brightest discreet sources have flux densities below the detection limit of microwave sensors in a single scan.

## 1   Introduction

Understanding and predicting climate variation requires accurate measurements of small changes over long periods of time. The global surface temperature, for example, has increased by only

$0.113°C$ decade$^{-1}$ over the period 1950 - 1999 (Karl et al. , 2015). Characterizing such trends from





space with sufficient accuracy is difficult, because the typical operational lifetime of a satellite rarely exceeds ten years, and the performance of an instrument might change in unexpected ways in orbit.

Previous studies to intercalibrate microwave sensors on different satellites have concentrated on
averaging Earth view data in certain ways or in finding collocations between measurements (e.g., John et al. , 2013, 2012). The top of the atmosphere of the Earth, however, is not an absolute calibration target, since it will change over time, especially in a changing climate. Hence there is a need for a nondegrading radiometric standard as reference for checking the stability of the absolute photometric calibration. The object best suited for this purpose is the Moon, which appears occasionally in the deep space view (DSV) of instruments on satellites for Earth observation on polar
orbits. As its spectral hemispherical emissivity is constant for all practical purposes (Kieffer , 1997), only the orbital effects, i.e. the Sun–satellite–Moon angle, the distances between Earth, Moon and Sun, and libration, can make a difference to the flux arriving in the DSV. After characterizing these variations by taking lunar images for several years with an automated observatory on ground - the
ROLO (RObotic Lunar Observatory) telescope (Kieffer and Stone , 2005) - the Moon was established as radiance standard for wavelengths between 0.35 and 0.95 $\mu$m (Kieffer and Wildey , 1996). It could be shown that nine lunar comparisons spread over one year are sufficient to detect changes, for example of the VIIRS reflective solar bands, at the 0.1% level (Patt et al. , 2005).

The ROLO project cannot be extended to longer wavelengths - thermal infrared or microwave - because the brightness of the stars needed for monitoring the calibration of the camera observing the Moon decreases proportional to the square of the wavelength, and the atmosphere is only transparent in certain spectral windows. It is possible, however, to correct for the orbital variations of the Moon's brightness considerably by fitting second order polynomials of certain orbital variables to
the radiances measured with the sensor itself. This method was used to obtain a model for subtracting the lunar contamination in the calibration of AMSU-A (Advanced Microwave Sounding Unit-A, Mo and Kigawa , 2007).

Adjusting for orbital effects purely on the basis of the observations with the satellite itself, how-
ever, requires a large dataset. Mo and Kigawa (2007) had to put up with unplanned lunar intrusions in the DSV of AMSU-A, which crossed the DSV at quite different distances from its center. It seems therefore desirable to establish a fixed pattern of spacecraft roll maneuvers for future missions like ICI (Ice Cloud Imaging) or MWI (MicroWave Imaging) for MetOp Second Generation (MetOp-SG) in order to measure their lifetime stability by means of observations of the Moon. In the following
we make the case for this calibration strategy.



## 2  Frequency of the Moon appearing in the deep space view

All microwave sensors are calibrated in flight by measuring the signal from both the Cosmoloical Microwave Background (CMB) at $2.72548 \pm 0.00057$ K (without foreground emission, Fixsen , 2009) and an on-board calibration target at some 280 K. The DSV points at a fixed angle $\alpha$ from nadir
in the anti-Sun direction and perpendicular to the flight direction (see Figure 1). $\alpha$ amounts to $65°$–$81°$ for AMSU-B (Charlton et al. , 1993) and to $71°$–$76°$ for MHS (Microwave Humidity Sounder, Goodrum et al. , 2014). In the course of one orbit of the satellite, the deep space viewing direction describes a circle with an opening angle $180° - 2\alpha$ in the sky. The center of this circle is about $8°$, i.e. the north-south orbital inclination of the satellite, away from the equator. The orbit of the
Moon around the Earth, however, is tilted by up to $28.6°$ against the celestial equator (the sum of the inclination of the Earth's axis of $23.45°$ against the ecliptic plane, and the Moon's orbital inclination of $5.15°$). Hence there are periods, when the Moon's orbit does not cross the circle described in the sky by the DSV direction during one orbit of the satellite. During these times, the Moon cannot be seen by the microwave sensor. They become longer with increasing $\alpha$: If the DSV was perpendicular
to nadir, the Moon would be visible at most twice per year, viz. when the orbit axis of the satellite crosses the Moon's orbit. On the other hand the Moon appears every month in the DSV, when $\alpha \leq 53.4°$, because in this case it must cross the DSV ring even at its largest possible distance from the celestial equator. MHS typically has the Moon within its DSV ring eight times a year (see Table 1). Usually there are two groups of contaminated orbits per month, which are separted by one
or two days, corresponding to the Moon's entry and exit in the DSV ring.

## 3  Uncertainties of the reference flux density from the Moon

Even though the Moon appears several times per year in the DSV of microwave sensors, and models are available for subtracting the contribution of this natural satellite to the reference flux from the CMB, this object has not been used as calibration standard at mm-wavelengths yet. The main reasons
are imperfections of the calculations of the temperature of the Moon and systematic errors caused by the fact that the Moon fills only a fraction of the field of view (FOV) of the instrument.

### 3.1  Radiant flux of the Moon

The algorithm used by Mo and Kigawa  (2007) for detection and correction of the lunar contamination in AMSU-A, which was later modified by Yang and Weng  (2016) for ATMS (Advanced
Technology Microwave Sounder), calculates the brightness of the Moon under the assumptions that the disk-integrated emissivity of the surface is 0.95, independent of the apparent "geodetic" longitude and latitude of the center of the Moon seen by the sensor, and that the disk-integrated temperature of the Moon is a quadratic function of $\cos \Theta$, $\Theta$ being the Sun–satellite–Moon angle. This picture could not make allowance for the complex structure of the lunar surface that was recently revealed at fre-



quencies similar to those of channels 1 and 2 of AMSU-A by China's First Lunar Probe Chang'E-1
(Zheng et al. , 2012). Therefore a comparison of the cold calibration count anomalies as predicted by
the lunar model with the observed ones showed considerable discrepancies: The measured contam-
ination for certain channels is up to 30% higher than the model value (see Mo and Kigawa , 2007,
Fig. 5). The agreement is better for ATMS, but also in this case there are systematic differences

between model and observations so that even after the removal of the lunar contamination small
gain variations remain (see Yang and Weng , 2016, Fig. 8). A more sophisticated model of the mi-
crowave radiation from the Moon, however, would require well-calibrated observations of the Moon
for years, in regular intervals and at a variety of frequencies, just like they have been carried out at
visible wavelengths. As the Earth's atmosphere is for the most part transparent to microwaves, such

an observing program is in principle feasible, but has to our best knowledge not been undertaken yet.

The range of possible angles $\Theta$ can be estimated according to the following formula. It is not quite
accurate, because the orbit of the Moon is tilted against the ecliptic. Obviously $\Theta$ cannot be larger
than $180°$.

$$\Theta_{min} = (12 - ECT) \cdot 15° + \alpha \tag{1}$$

$$\Theta_{max} = (12 - ECT) \cdot 15° + 180° - \alpha \leq 180° \tag{2}$$

where
ECT is the absolute value of difference of equator crossing time and noon in hours and is always
less or equal to six.

$\Theta$ follows from the angle between the direction of the Sun and the orbital plane of the satellite
and the angle between the direction of the DSV, pointing at the Moon, and the orbital plane of the
satellite. As the equator crossing time drifts during the lifetime of most satellites, the phase angles
of the Moon during an intrusion in the DSV vary as well betweeen start and end of mission.

### 3.2 Uncertainties of the antenna pattern and the Moon's position in the beam

As the nominal 3dB beamwidth is $3.3°$ for AMSU-A and $1.1°$ for AMSU-B and MHS, the full disk
of the Moon with an equatorial angular width between 29.4' and 33.5' fills only part of the DSV.
It is therefore very important to know exactly where the Moon is relative to the center of the beam.
This angle is supposed to be known with an accuracy of $0.3°$ according to the MHS Level 1 Product
Generation Specification. The JPL Horizons web interface, however, provides right ascension and

declination with an accuracy better than one arcsecond. These coordinates have to be transformed
to the position of the spacecraft and to the instrument FOV reference frame. In addition to the error
in the Moon calculation one has to consider the line of sight offset of the DSV. With AMSU-B, for
example, it is specified to have a knowledge of the pointing of the antenna of $\pm 0.05°$, i. e. negligible
compared to the assumed error in the position of the Moon (Charlton and Jarrett , 1993). Assuming





that the antenna beam patterns are approximately Gaussian, a distance from the center of $0.3°$ trans-
lates to a loss of antenna efficiency of 2% for AMSU-A and 20% for AMSU-B and MHS with their
smaller beams. As the Moon is an extended source, these values are lower limits for the actual loss in
signal. The 3 dB beamwidth itself is only known with an accuracy of 10% for AMSU-B (Atkinson ,
2001). This means an additional loss in signal of 14%, if a point source is actually 10% further away

from the center of the beam than the assumed 3dB beamwidth.

Figure 2 shows the light curve of the Moon in different DSVs for a particular orbit. Its minimum
distance from the center of the FOV is different for each DSV, because they point in slightly different
directions. The maximum signal is reached at closest approach, so the time difference between the

minima in the upper panel of Fig. 2 and the maxima in the lower panel is indicative of the actual
pointing error of the satellite. It amounts to some 10 sec for DSV 2 in Fig. 2, this is among the larger
values we could find from examining a few dozen intrusions of the Moon. From the angular velocity
of the DSV direction in the sky, $\omega$, we obtain the typical pointing error of MHS.

$$\omega = 2\pi \cdot (90° - \alpha) / 6000 \, sec \approx 0.017° \, / \, sec \qquad (3)$$

This means that the error in the Moon calculation is at most $0.17°$, i. e. smaller than assumed in the
MHS Level 1 Product Generation Specification. The movement of the Moon itself with an angular
velocity of $1.4° \times 10^{-4} / \, sec$ was neglected.

## 4   Benefits from measuring the Moon signal during a pitchover maneuver

### 4.1   Reduction of errors and uncertainties

The uncertainties described in Sect. 3.2 can be greatly reduced, if dedicated observations of the
Moon are used to check the stability of the calibration of microwave sensors, because they are car-
ried out at the same phase angle and with the Moon in the center of the beam. The exact position of
the Moon can be determined with a small raster map, if necessary.

Flux variations due to changes in distance between the sensor and the Moon can be corrected
for with an inverse square law, those due to changes in distance between the Sun and the Moon,
however, are more difficult to handle. They can be estimated with the Stefan-Boltzmann Law:

$$d_{SM}^{-2} \propto F_{in} = F_{out} \propto T_{eff}^4 \qquad (4)$$

where

$F_{in}$ is the flux absorbed by the Moon

$F_{out}$ is the flux emitted by the Moon

$d_{SM}$ is the distance between the Sun and the Moon



$T_{eff}$ is the effective temperature of the Moon

Flux changes due to variations of longitude and latitude of the center of the lunar disk are difficult
to characterize, because the periods of the two kinds of libration are slightly different, and either is
different from the synodic month. The maximum libration is 7°53' in longitude and 6°40' in latitude.
As the strongest longitudinal variations in microwave emission are found near $\lambda = 0°$ (Zheng et al.
, 2012), however, the effect that libration has on the microwave brightness is negligible compared to
the one of changing phase angle. The situation is similar in the thermal infrared, where libration in
latitude causes a systematic error of less than 1% (Daniels et al. , 2015). Unlike random appearances
in the DSV, pointing the instrument at the Moon only at a certain phase with dedicated maneuvers
of the satellite therefore eliminates the dominating component of the Moon's flux variations.

### 4.2 Lifetime stability

The main aim of observations of the Moon is a check of the stability of the photometric calibration.
Its standard procedure converts Earth view counts to radiance by the aid of repeated viewings of
the on-board black body target and cold space. The assumed flux from the former, however, can be
affected by time-dependent systematic errors, e. g. inaccurate thermometers or changing emissivity.
It is therefore desirable to establish a "reference for the reference" that is not affected by wear and
tear. This "higher order" standard should be observed at least at the start and the end of the mission
in order to detect monotonous drifts. The higher the number of such spacecraft maneuvers, the better
the characterization of the stability of the sensor. Therefore lunar observations were carried out some
ten times per month at or near opposition with the Clouds and the Earth's Radiant Energy System
(CERES) in order to provide enough data for reducing the remaining uncertainties, e. g. those caused
by libration. Daniels et al. (2015) were then able to detect trends of 1% per decade or less in the
infrared channels of CERES.

### 4.3 Examples: Ice Cloud Imager and Microwave Imager on EPS-SG

There will be two microwave imaging radiometers for MetOp-SG: ICI and MWI. Unlike AMSU-B
and MHS they will carry out conical instead of cross-track scans (Alberti et al. , 2012). This detail
makes lunar observations with pitchover maneuvers even more useful, because they will allow to
observe both deep space and the Moon without the cold calibration reflector, whose optical properties
introduce an additional error source in the routine calibration of every scan. With a small raster
map, it will also be possible to measure the antenna pattern in flight and to monitor its stability.
The accuracy requirement of these instruments is 0.5 K, and it will be easy to bring the random
uncertainty far below this value by combining a sufficient number of exposures of the Moon. The
instantaneous  FOV is smaller than that of the precursor instrument MHS - 0.65° versus 1.11° - so
one can expect a higher signal from the Moon compared to microwave background and blackbody.
0.65° are still sufficient to cover the full disk of the Moon even at its smallest distance from Earth,



but they require a pointing accuracy of (0.65° - 0.5°) / 2 = 0.075° to make sure the Moon is within the borders of the 3 db-beamwidth. Hence the step size of the above mentioned raster map should not be larger than a few arcmin.

## 5   Other intruders in the deep space view

The Moon is the brightest solar system object except for the Sun. Mo and Kigawa (2007) estimate its brightness temperature to be 258 K at all frequencies for $\Theta = 127.5°$, a typical value for Moon intrusions with MHS on MetOpA and MetOpB. This means that the increase of counts from the DSV caused by the intrusion of the Moon is a bit less than the fraction of the FOV covered by the Moon multiplied by the count difference between DSV and black body level (see Table 2).

Table 2 contains also entries for two other sources: Jupiter, brightest radio source among the outer planets, and the Crab Nebula, a supernova remnant that strongly emits synchrotron radiation. It can be seen that for the currently operational microwave satellites neither object poses a problem in terms of contamination of the DSV: At 23.8 GHz and a FOV with 3.3° diameter, the contribution is smaller than the resolution of the analog-to-digital converter, at 183.3 GHz and a FOV with 1.1° diameter, the contribution is about an order of magnitude smaller than the noise. This might change with ICI and MWI, however, because their FOVs are smaller than for previous sensors, and therefore the brightness of any source entering the DSV relative to the black body signal increases.

## 6   Conclusions

With decreasing beamwidth at 183 GHz - 3.3° for SSM/T-2, 1.1° for AMSU-B, and 0.65° for MWI - the signal of the Moon at its appearances in the deep space view has increased relative to the contribution from the cosmic microwave background. As a consequence, it will be possible for future missions to measure the lunar flux with sufficient accuracy to check whether the requirements on the stability of the instrument are fulfilled. While the emissivity of the Moon itself can confidently assumed to be constant, it is essential to minimize the uncertainties related to the dependence of the flux on phase angle and the position of the Moon in the field of view. Both sources of error are best addressed with dedicated spacecraft maneuvers. If such maneuvers include also a raster map, one can characterize the beam pattern in flight. For this purpose, however, deconvolution of the images will become necessary, because the Moon is an extended source, especially for ICI and MWI. A raster map would also aid to determine the exact position of the Moon. If an accuracy of a tenth of the 3 db bandwidth could be achieved, the corresponding flux error would amount to only a few percent.



Its invariability makes the Moon also well suited for intercalibration between sensors that were operational at quite different epochs. Given the fact that the unplanned lunar intrusions in the deep space view happen over a wide range of phase angles, because of orbital drifts and different viewing geometries, it is quite possible to find pairs of observations with similar phase angles from different satellites. This way one eliminates the unavoidable differences in scene temperature that adversely affect other methods like simultaneous nadir overpasses and zonal averages. A ground-based observing program of the Moon at mm-wavelengths would considerably reduce the errors associated with remaining phase angle differences and libration. Microwave sensors in space, however, do not only operate at wavelengths, for which the Earth's atmosphere is transparent, and Zheng et al. (2012) showed that the brightness temperature of the warmest features on the lunar surface can vary by 30 K between different frequencies. Hence a sophisticated model of the brightness of the Moon must allow for such variations with frequency. This requires some measurements to be carried out from an airplane flying in the stratosphere, e. g. ISMAR (International SubMillimeter Airborne Radiometer, Fox et al. , 2014).

While the Moon produces thousands of counts when moving through the deep space view, the possible impact from other discreet sources in the sky on the low level reference flux from the cosmic microwave background is orders of magnitude smaller. Even with future instruments, intrusions in the DSV from objects other than the Moon will have no significant impact on the calibration accuracy.

Author contributions  S. Buehler started and guided this investigation. I. Hans, T. Lang, and S. Michel developed the MATLAB code for reading the MHS Level 1b Science Packets. T. Lang and S. Michel identified instances of intrusions of the Moon in the deep space views of the microwave sensors. M. Burgdorf prepared the manuscript with contributions from all co-authors.

*Acknowledgements.* This work was undertaken within the project "Fidelity and Uncertainty in Climate data records from Earth Observation (FIDUCEO)". FIDUCEO has received funding from the European Union's H2020 Research and Innovation programme, under Grant Agreement 638822. The authors wish to thank Nigel Atkinson and Oliver Lemke for helpful tips on using the ATOVS and AVHRR Pre-processing Package.



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





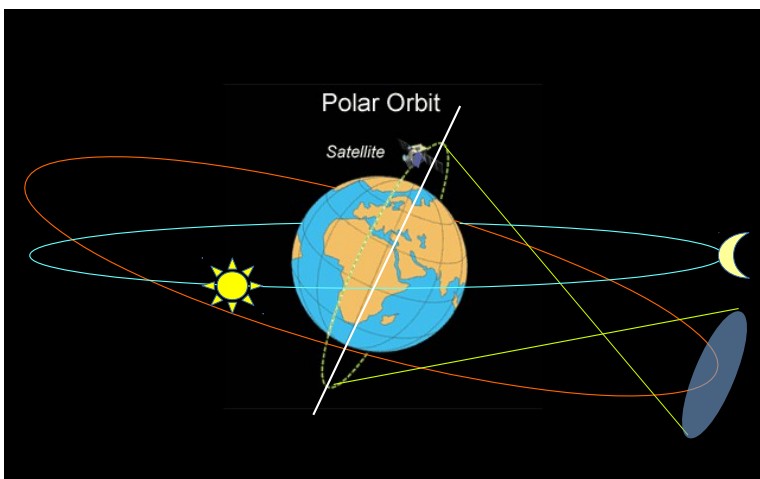

**Figure 1.** Viewing direction of the Deep Space View (DSV, green) compared to the Celestial Equator (red) and the ecliptic plane (blue). For simplicity, the slight tilt of the Moon's orbit against the ecliptic and the tilt of the orbital axis of the artificial satellite against the equator are not displayed. The DSV direction has an angle $\alpha$ against nadir and describes a circle in the sky during one orbit (grey shaded area).





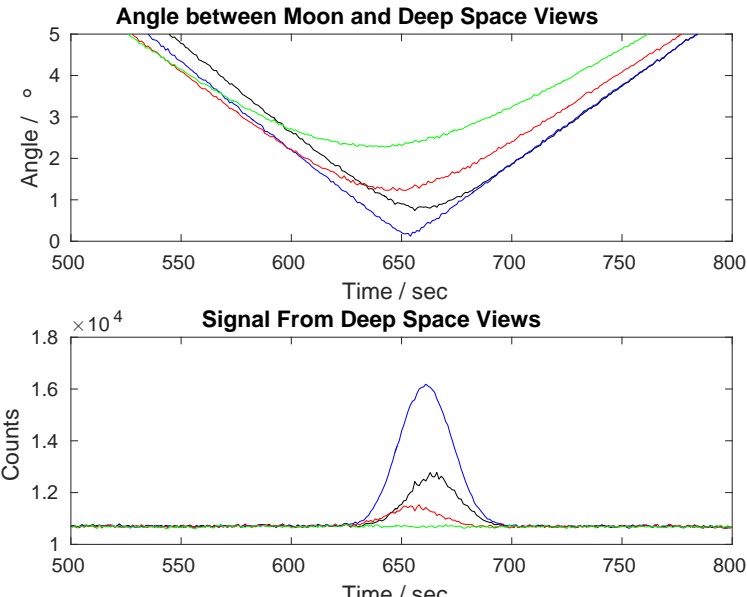

**Figure 2.** Moon intrusion event in different deep space views (1: black, 2: blue, 3: red, 4: green) on 10/30, 2015, with MHS (Microwave Humidity Sounder) on MetOpA.



**Table 1.** Frequency of appearances of the Moon in at least one of the four positions of the deep space view. The fourth column gives the range of angles between the orbital axis of the artificial satellite and the four DSV positions. Intrusions affect between six and 36 orbits in a row; the fifth column gives the months, in which they happened. There is usually not more than 1% of lunar contaminated scans per orbit of MHS. Up to a third of all scans in one orbit, however, can be affected by the Moon with AMSU-A, which is included in the table as an example for a different instrument. $\Theta$ is the Sun–satellite–Moon angle.

| Satellite | Instrument | Begin Utilization | Orbital Axis - DSV | Intrusions 2015 | $\Theta$ |
|---|---|---|---|---|---|
| MetOp-B | MHS | 09/24/2012 | $14.2° - 18.6°$ | 1–2, 7–12 | $127.5° \pm 18.6°$ |
| NOAA-19 | MHS | 02/07/2009 | $14.2° - 18.6°$ | 1–5, 10–12 | $100° - 140°$ |
| MetOp-A | MHS | 01/01/2007 | $14.2° - 18.6°$ | 1–2, 7–12 | $127.5° \pm 18.6°$ |
| NOAA-18 | MHS | 05/24/2005 | $14.2° - 18.6°$ | 1–5, 9–12 | $100° - 180°$ |
| NOAA-15 | AMSU-A | 10/26/1998 | $6.7° - 13.3°$ | 2–3, 9–11 | $140° - 180°$ |





**Table 2.** Brightness of different astronomical sources. No brightness temperatures are given for the Crab Nebula, for this is not thermal emission. The flux densities $\Phi$ in the fifth and sixth column are normalized to the black body. The values were calculated for two channels with very different frequencies on AMSU-A and MHS; the exposures of DSV and black body differ typically by 3500 counts for channel 1 of AMSU-A and 47,000 counts for channel 3 of MHS.

| Object | $T_B^{23.8\ GHz}$ | $T_B^{183.3\ GHz}$ | Diameter | $\Phi$ at 24 GHz | $\Phi$ at 183 GHz | Ref. |
|---|---|---|---|---|---|---|
| BB | 280 K | 280 K | — | 1 | 1 | |
| CMB | 2.72548 K | 2.72548 K | — | $7.9 \times 10^{-3}$ | $1.3 \times 10^{-3}$ | Fixsen (2009) |
| Moon | 258 K | 258 K | 30 arcmin | $2.1 \times 10^{-2}$ | $1.9 \times 10^{-1}$ | Mo and Kigawa (2007) |
| Jupiter | 130 K | 150 K | 0.8 arcmin | $7.6 \times 10^{-6}$ | $7.8 \times 10^{-5}$ | de Pater and Lissauer (2000) |
| Crab Nebula | S = 400 Jy | S = 290 Jy | 10 arcmin | $3.2 \times 10^{-5}$ | $2.3 \times 10^{-6}$ | Macias-Perez et al. (2010), Mezger et al. (1986) |