# Peer review of "The Moon as a photometric calibration standard for microwave sounders"

_Atmospheric Measurement Techniques, 2016_

## Referee Comment (RC1) · Anonymous Referee #1 · 18 Mar 2016

This paper discussed the possibility to use Moon as calibration standard but no technique details were presented. A few aspects related to determine the Moon radiance are discussed in the paper. However, the current manuscript has a great deal of blunt statements which are scientifically un-justified. At this stage the paper can only be accepted after a major correction, proving to be valid, and revision based on above comments. Overall, I hope the authors would provide more technique detail based on data facts as well as improve their writing of this paper. The manuscript requires substantial revisions prior considerations for publications. Hence, I will give my specific comments to guide the revisions expected from authors.

section 2, line 70 to 80. The author claimed that the frequency of lunar intrusion to deep space view is decrease with space view angle increase. It is better to have some simulation/observation results for existing microwave instruments like AMSU/MHS/ATMS

section 3.1. The radiance of Moon is a function of Moon phase, which is defined as angle between viewing vector of Moon and the Sun from satellite at any time DSV observations. Since there are already plenty of lunar intrusion cases during AMSU/MHS life time, the DSV data with LI under different Moon phase can be collected. The author should use these datasets to do some study on Moon radiance to support the statement in this section

section 4.1. variation of distance between the sensor and the Moon will impact the the normalized solid angle, which is defined as an area ratio of the full disk of the Moon and the antenna response. To determine the Moon radiance, solid angle is need to be known accurately. An analysis for impact of solid angle on Moon radiance should be added in the study
* * *

---

## Author Comment (AC1) · 24 Mar 2016

This is the author's reply on the comments by referee #1.

Leaving aside the question whether some statements in our manuscript are "blunt" and "unjustified" for the simple reason that I do not know which ones these are supposed to be, I shall address the more specific comments only.

1. Frequency of lunar intrusion in the deep space view: I welcome the suggestion by the referee to provide results for existing microwave instruments. They will be calculated for AMSU/MHS/ATMS and included in the manuscript.

2. Radiance of the Moon as a function of its phase: The referee is right that this function is not known with an accuracy sufficient for absolute flux calibration of AMSU-

[Figure]

B or MHS. Previous estimations of the lunar contamination in the deep space views of AMSU-A and ATMS (Yang & Weng, Mo & Kigawa) only aimed at correcting these DSV reference measurements. Therefore in the microwave range the Moon can at best be used for checking the stability of the flux calibration by comparing observations at the same phase. This will be made clearer in the manuscript. The development of a lunar model that is more sophisticated than those published already (see above) is beyond the scope of the manuscript submitted. The suggestion by the referee to do at least "some study on Moon radiance", however, is understood mainly as a request for additional support of statements made on the lunar flux and its dependencies, and we shall have it in the revised version of the manuscript. A "final model" taking libration, frequency and spatial dependence of emissivity, etc. into account will make a good topic for a future publication.

3. Accuracy of normalised solid angle: The main source of uncertainty in the normalised solid angle of the Moon is the error in the beam width of the antenna. This is more relevant for an absolute calibration with the Moon than for just checking the stability of the calibration (see previous paragraph), but we accept the suggestion to add a more detailed discussion of this point to the manuscript.

---

## Referee Comment (RC2) · Anonymous Referee #2 · 15 Apr 2016

Overall:

I find the subject matter of the manuscript interesting and of value to the community. However, I feel the manuscript is lacking in any detailed analysis to back up the author's claims. Perhaps the intent of the manuscript is just to present a theoretical analysis, but I think it would greatly add to the paper to show some current data, since there are several microwave sensors currently in orbit that the authors can use to show the use of the moon as a calibration target.

Specific Comments:

It's not clear to me in the paper just how accurate the authors expect the moon as a calibration target to be. I would like to see them estimate some accuracy in terms of Kelvins. It seems like there is a lot of error associated with using the moon as a

calibration target, and it's unclear to me what value a moon calibration target would add in addition to using deep space maneuvers. Dedicated maneuvers of spacecrafts have been performed to view deep space (e.g. TMI and GMI), so it would be worthwhile to mention what benefit of doing a maneuver to view the moon would add, since deep space is much more stable, is an accurately known value, and easier to measure than the moon as a calibration target. The authors state the accuracy of ICI and MWI is 0.5 K and say "it will be easy to bring the random uncertainty far below this value by combining a sufficient number of exposures to the moon", but I didn't see anything to back up that claim.

Throughout the paper there are references to 'microwave sensors'. The title states using 'microwave sounders'. The authors mention AMSU-A, MHS, and AMSU-B which are all sounders, but then they say this can be used for the future ICI and MWI, which are not sounders. It would be good to clarify the language of exactly which microwave sensors you think the moon calibration target can be used for, including the frequency range since these sensors encompass a very wide range of frequencies.

It would be helpful for the reader who is not as familiar with the microwave sensors you describe to include a table of at least the frequencies for AMSU-B/A and MHS to help understand which frequency range you are talking about.

As stated above, I feel the manuscript would be enhanced by showing some examples of what the moon looks like in the DSV. Since it sounds like no dedicated spacecraft maneuver has been done to view the moon as you proposed, some examples of how it appears in the DSV may help the reader with understanding what the moon looks like in the DSV and allow the authors a way to validate their claims.

Page 2, line 31. The top of the atmosphere of the Earth is not used as a calibration target. Perhaps you mean TOA brightness temperature? I would re-word to be more accurate.

Page 3, first sentence says "All microwave sensors..." but this statement is incorrect

since not all microwave sensors use a 2 point calibration.

---

## Author Comment (AC2) · 26 Apr 2016

This is the author's reply on the comments by referee #2.

The comment on the value of the manuscript is greatly appreciated.

Concerning the first specific comment, I agree with the referee that there is a large error to the Moon's absolute brightness temperature in the microwave range. We know, however, that its emissivity does not change with time. Therefore it is not well suited as absolute flux standard and should rather be used to check the stability of the flux calibration or to perform inter-calibration. This cannot be done with the Cosmic Microwave Background alone. I realise that this point has to be made clearer in the manuscript and the discussion of the accuracy that can be achieved will be expanded. The suggestion by the referee to include more examples of the Moon in the DSV is very helpful

in this respect and will be followed.

All other comments are gratefully noted and will be reflected in the next version of the manuscript.
* * *